# Economic evaluation of trastuzumab in HER2-positive early breast cancer in Indonesia: A cost-effectiveness analysis

Sudewi Mukaromah Khoirunnisa[1,2]*, Fithria Dyah Ayu Suryanegara[1,3], Didik Setiawan[4,5], Maarten Jacobus Postma[1,6,7,8], Lisa Aniek de Jong[1]

1 Department of Health Sciences, University of Groningen, University Medical Center Groningen, Groningen, The Netherlands, 2 Department of Pharmacy, Institut Teknologi Sumatera, Lampung Selatan, Indonesia, 3 Department of Pharmacy, Universitas Islam Indonesia, Yogyakarta, Indonesia, 4 Faculty of Pharmacy, Universitas Muhammadiyah Purwokerto, Banyumas, Indonesia, 5 Centre for Health Economic Studies, Universitas Muhammadiyah Purwokerto, Banyumas, Indonesia, 6 Faculty of Economics & Business, Department of Economics, Econometrics and Finance, University of Groningen, Groningen, The Netherlands, 7 Faculty of Medicine, Department of Pharmacology and Therapy, Universitas Airlangga, Surabaya, Indonesia, 8 Centre of Excellence in Higher Education for Pharmaceutical Care Innovation, Universitas Padjadjaran, Bandung, Indonesia

* s.m.khoirunnisa@rug.nl, sudewi.mukaromah@fa.itera.ac.id

**Data Availability Statement:** Due to legal constraints, as this study involves third-party data (BPJS Kesehatan), the authors cannot legally

## Abstract

### Background

Trastuzumab has significantly enhanced the survival and prognosis of individuals diagnosed with human epidermal growth factor receptor 2 (HER2)-positive early breast cancer. Considering its relatively high costs, we aimed to examine the cost-effectiveness of trastuzumab plus chemotherapy compared with chemotherapy alone in HER2-positive early breast cancer from an Indonesian healthcare payer's perspective.

### Methods

A Markov model was developed to project the lifetime health benefits and costs associated with trastuzumab treatment for a cohort of women with HER2-positive early breast cancer. Efficacy data and baseline characteristics in the base-case analysis were primarily derived from the 11-year results of the HERA trial. Costs were based on verified reimbursement data from Indonesia's Health and Social Security Agency (BPJS Kesehatan) of the year 2020. A scenario analysis was conducted with efficacy data based on the joint analysis from the NSABP B-31 and NCCTG N9831 trials, allowing for subgroup analysis by age at diagnosis. Univariate and probabilistic sensitivity analyses were conducted to assess the influence of parameter uncertainty.

### Results

In the base-case analysis, the results indicated that the lifetime costs for trastuzumab plus chemotherapy and chemotherapy alone were US$33,744 and US$22,720, respectively, resulting in substantial incremental savings of US$11,024 per patient for the former.

distribute it. The data supporting the findings of this study are publicly available. We would like to inform other researchers interested in accessing the data that they can submit a request through the JKN Portal available at https://data.bpjs-kesehatan.go.id/ (more information is in the Supplementary material).

**Funding:** This study was supported through Lembaga Pengelola Dana Pendidikan (LPDP) under grant number 0005479/PHA/D/BUDI-2019 and by the University of Groningen/University Medical Center Groningen. The funders had no role in study design, data collection and analysis, decision to publish, or preparation of the manuscript.

**Competing interests:** The authors declare that they have no competing interests

Trastuzumab plus chemotherapy also led to higher total quality-adjusted life years (QALYs) and life years gained (LYG), resulting in incremental cost-effectiveness ratios (ICERs) of US $6,842 per QALY and US$5,510 per LYG. In scenario analysis, the subgroup with an age at diagnosis <40 years old reflected the most cost-effective subgroup. Both the base-case and scenario analyses demonstrated cost-effectiveness with a willingness-to-pay threshold of three-times Gross Domestic Product (GDP). Sensitivity analyses confirmed the robustness of the findings and conclusions.

## Conclusion

In Indonesia, trastuzumab plus chemotherapy can be considered cost-effective compared to chemotherapy alone at a willingness-to-pay threshold of three times GDP, and it is likely most cost-effective in women <40 years of age.

## Introduction

Breast cancer currently occupies the leading position as the most prevalent cancer globally [1]. The most common subtype is human epidermal growth factor receptor 2 (HER2)-positive breast cancer, constituting 20–30% of cases with an age-adjusted incidence rate of 87.2% in 2020 [1]. HER2-positive breast cancer patients typically exhibit a more aggressive tumor with high resistance to chemotherapy, leading to a poorer prognosis [2].

The HER2-targeted drug trastuzumab has significantly improved the survival and prognosis of individuals diagnosed with HER2-positive breast cancer [3–5]. Adding trastuzumab to chemotherapy for early-stage HER2-positive breast cancer results in a 30% reduction in breast cancer recurrence and mortality observed across various patient and tumor characteristics, making trastuzumab a valuable treatment strategy [6].

The final analysis of the HERceptin Adjuvant (HERA) trial, with a median follow-up of 11 years, suggests that administration of trastuzumab for one year after chemotherapy in individuals diagnosed with HER2-positive early breast cancer demonstrates significant improvement in long-term disease-free survival (DFS) [5]. Since breast cancer tumors in younger women are often more likely to be hereditary, larger in size and grade, and often lack estrogen receptor and progesterone receptor expression, which is related to unfavorable distant recurrence and overall survival (OS) [7], it is essential to consider heterogeneity in terms of age at diagnosis. A joint analysis of NSABP B-31 and NCCTG N9831 evaluating the efficacy and safety of trastuzumab by age group found that patients younger than 40 years receiving trastuzumab plus chemotherapy had better DFS and OS compared to the older age groups [3].

Based on oncological consensus, the recommended treatment duration for trastuzumab in Asian countries is similar to that of the United States and Europe [8]. The optimal treatment duration for trastuzumab therapy is one year, preferably in combination with an anthracycline-based chemotherapy regimen [8]. However, despite the guideline recommendation, certain (Asian) low- and middle-income countries, including Indonesia, have not been able to provide the complete the full one-year treatment course due to economic constraints [9,10].

Given the potentially high acquisition costs of trastuzumab, evaluating its economic impact and health benefits is crucial. Economic evaluations play a vital role in enabling informed decision-making about the use of breast cancer treatments, ideally aiming to optimize patient outcomes and minimize overall costs. Although economic studies of trastuzumab have been

conducted in Western countries [11–14], evidence in Asian regions is still limited. We aimed to investigate the cost-effectiveness of trastuzumab plus chemotherapy compared with chemotherapy alone for HER-positive early breast cancer in Indonesia from a healthcare payer's perspective. In addition, we aimed to perform subgroup analysis based on age at diagnosis to inform potential clinical decision making.

## Materials and methods

### Model

An economic evaluation was conducted to estimate the cost-effectiveness of trastuzumab plus chemotherapy compared with chemotherapy alone in HER2-positive early breast cancer patients in Indonesia. A Markov model was developed, consisting of four health states [12,15]: 1) disease-free, 2) locoregional recurrence, 3) metastasis, and 4) death, with a cycle length of one year (Fig 1). All patients entered the model in the disease-free state, after which they could transition to the other states or remain in the disease-free state. Patients could only transition to the death state from the locoregional recurrence or metastatic state. The death state was an absorbing health state. The time horizon of the model was 50 years (lifetime) and the analysis was conducted from a healthcare payer's perspective.

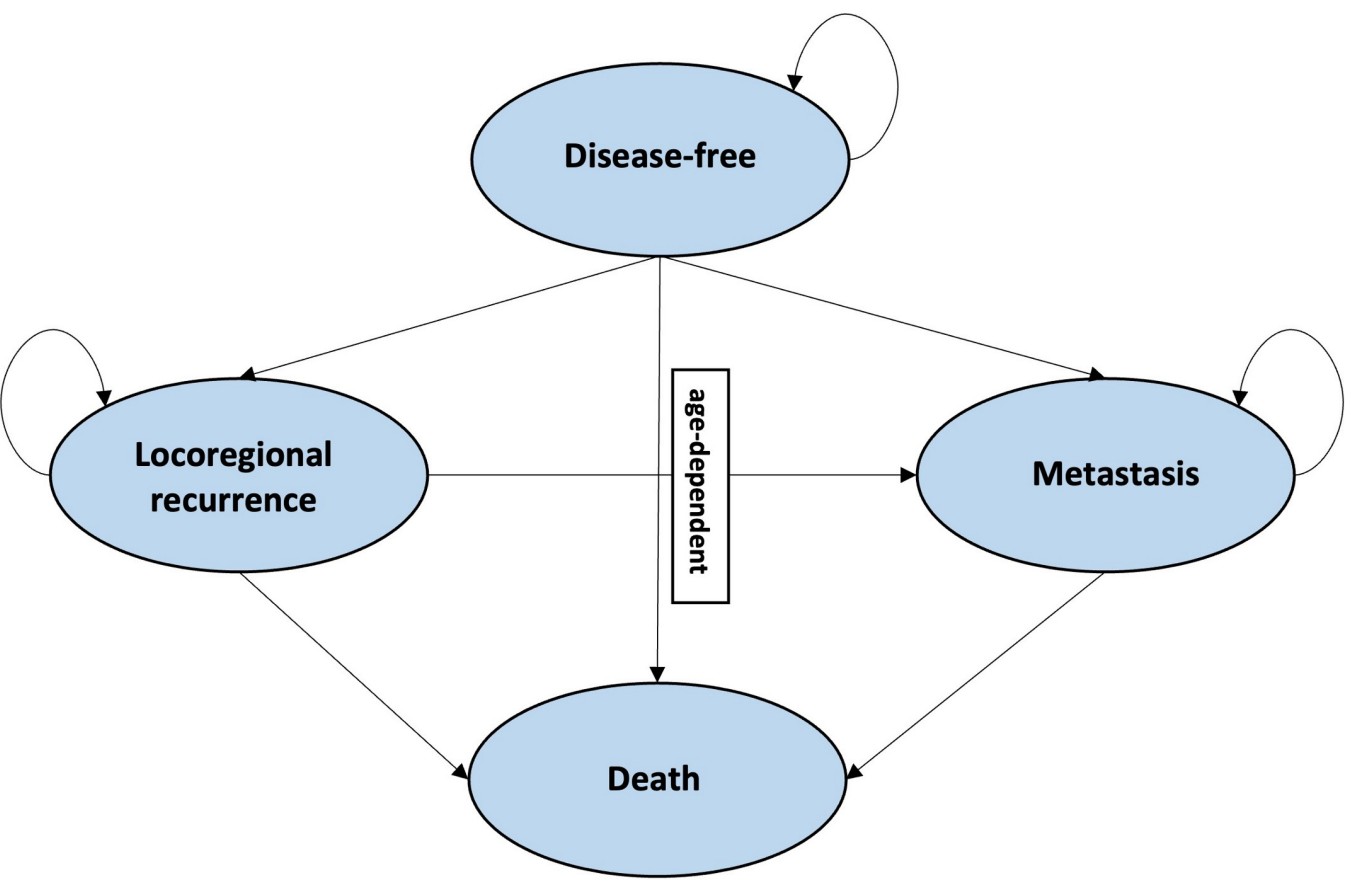

**Fig 1. Schematic representation of the Markov model.**

## Patient population and treatment

The baseline characteristics of the hypothetical cohort were based on the pivotal HERA trial [4,5,16]. The cohort consisted of women with a median age of 49 years with HER2-positive early breast cancer. In accordance with the HERA trial, patients received an anthracycline with a taxane-based chemotherapy regimen, which is consistent with standard therapy in Indonesia. Trastuzumab was administered intravenously: a loading dose of 8mg/kg was followed by a maintenance dose of 6mg/kg every three weeks for one year. The WHO average Indonesian female weight (56.2 kg) [17] was used to calculate the mean dose of trastuzumab per patient (equal to 449.6 mg loading dose and 337.2 mg maintenance dose).

## Transition probabilities

In the base-case analysis, the annual transition probabilities and the hazard ratio (HRs) for trastuzumab plus chemotherapy versus chemotherapy alone for OS, DFS, and metastasis were derived from HERA trial with an 11-years follow-up [5]. The HERA trial was chosen for the base case analysis because this is the trial with the longest follow-up available. Age-dependent transition probabilities were taken into account for the transition from disease-free to death, using the mortality rate in Indonesia reported by the WHO [18] (Table 1). The treatment effect was assumed to be constant over the lifetime horizon.

## Utilities and costs

In the absence of consistent and complete data specific to Indonesia, utility values for each health state were based on a published study from Thailand [20]. Only direct medical costs were taken into account as the analysis was conducted from a healthcare payer's perspective, which includes treatment and health state costs. The treatment duration of trastuzumab in the disease-free state was one year, based on the HERA trial [4]. The cost of trastuzumab per vial (US$ 384/440mg vial) was derived from the National Public Procurement Agency (NPPA) Indonesia e-katalog [22]. The costs for health states, including those for disease management such as chemotherapy, radiotherapy, and surgery, were calculated based on unit costs obtained from verified reimbursement data provided by Indonesia's Health and Social Security Agency (BPJS Kesehatan) for the year 2020 [21]. The resource use for each of these disease management treatments was based on literature [24]. The Gross Domestic Product (GDP) deflator was used to adjust for inflation [25] and all costs were converted to 2023 US$ [26]. An overview of the model input parameters used in the analysis is presented in Table 1.

## Model outcomes

Total lifetime costs, life years, and quality-adjusted life years (QALYs) were calculated. The incremental cost-effectiveness ratio (ICER) was determined by dividing the incremental costs associated with trastuzumab plus chemotherapy and chemotherapy alone by the corresponding incremental QALYs and life years gained (LYG). Trastuzumab was considered cost-effective when the ICER fell below the willingness-to-pay (WTP) threshold of three times GDP per capita, which is recommended for Indonesia [27]. The 2022 GDP per capita for Indonesia was US$4,788 [23]. Half-cycle correction [28] was applied and both costs and utilities were discounted by 3%, in accordance with Indonesian guidelines for health economic evaluations [29]. All analyses were conducted using Microsoft® Excel.

**Table 1. Model input parameters.**

| Parameters | Estimation of the base-case | 95% CI | Distribution | Source |
|---|---|---|---|---|
| **Mortality rates** | | | | |
| **Probability of death from all causes** | Age-dependent mortality rates from WHO | - | Fixed | [18] |
| **Transition probability and treatment effect** | | | | |
| Disease-free survival to locoregional recurrence | 0.029 | 0.009 to 0.035 | Beta | [4,19] |
| Disease-free survival to metastasis | 0.078 | 0.039 to 0.106 | Beta | [4,19] |
| Disease-free survival to death | Age-dependent | - | Beta | - |
| Locoregional recurrence to metastasis | 0.078 | 0.039 to 0.106 | Beta | [4,19] |
| Locoregional recurrence to death | 0.019 | 0.016 to 0.024 | Beta | [4,15] |
| Metastasis to death | 0.728 | 0.725 to 0.731 | Beta | [4,19] |
| **Relative treatment effect (trastuzumab + chemo vs chemo alone)** | | | | |
| Hazard ratio Overall Survival | 0.740 | 0.640 to 0.860 | Log-normal | [5] |
| Hazard ratio Disease-free survival | 0.760 | 0.680 to 0.860 | Log-normal | [5] |
| Hazard ratio for distant metastasis | 0.740 | 0.640 to 0.860 | Log-normal | [5] |
| **Utilities** | | | | |
| Disease-free survival | 0.847 | 0.807 to 0.886 | Beta | [20] |
| Locoregional recurrence | 0.810 | 0.760 to 0.870 | Beta | [20] |
| Metastasis | 0.484 | 0.426 to 0.542 | Beta | [20] |
| Death | 0 | | Fixed | Assumption |
| **Health state cost, including disease management and chemotherapy (2023, US$)** | | | | |
| Disease free survival in 1st year | 9,080 | 7,264 to 10,896 | Gamma | Reimbursement cost from BPJS Kesehatan [21]* |
| Disease free survival | 2,550 | 2,040 to 3,060 | Gamma | Reimbursement cost from BPJS Kesehatan [21]* |
| Locoregional recurrence | 3,416 | 2,732 to 4,099 | Gamma | Reimbursement cost from BPJS Kesehatan [21]* |
| Metastatic | 3,356 | 2,685 to 4,027 | Gamma | Reimbursement cost from BPJS Kesehatan [21]* |
| **Intervention cost** | | | | |
| Trastuzumab - 17 cycles (1 year) | 6,530 | 5,749 to 7,311 | Gamma | [22] |
| **Other** | | | | |
| GDP per capita 2022 | 4,788 | - | Fixed | [23] |

Abbreviations: WHO, world health organization.

Note: Health state costs were based on reimbursement data from BPJS Kesehatan of woman diagnosed with breast cancer identified with ICD-10 code C50, and the verified cost was based on INA-CBG (Indonesia Case-Base Groups) codes reimbursed for this specific group.

## Scenario analysis

Several scenario analyses were carried out in this study. One scenario analysis was based on a treatment effect duration of 11-years, equivalent to the follow-up period of the HERA trial, to assess the impact of assuming a continuous treatment effect over a lifetime horizon in the base-case analysis. Additionally, taking into account that age at diagnosis is associated with the OS of breast cancer patients [7,30,31], a scenario analysis by age at diagnosis was performed. However, because the HERA trial [5] does not contain published hazard ratios (HRs) per age group, the HRs in this scenario were based on data from a joint analysis from NSABP B-31 and NCCTG N9831 [3]. This joint analysis compared trastuzumab plus chemotherapy with chemotherapy alone, with a median follow-up of 8.4 years [3]. Based on this analysis, four

additional scenarios were conducted based on age at diagnosis: <40, 40–49, 50–59, and ≥60 years old. The input parameters for scenario analysis are shown in Table 2.

### Sensitivity analysis

One-way sensitivity and probabilistic sensitivity analyses (PSA) were performed to evaluate the robustness of the model results. In the one-way sensitivity analysis, each parameter was varied separately within its respective 95% confidence interval (95% CI). If a 95% CI was not available, parameters were varied by +/-10% from their base-case values. A range of 0% to 5% was used for discount rates. The results of the one-way sensitivity analysis were presented in a tornado diagram.

In the PSA, distributions were assigned to each input parameter, drawing random values for all parameters simultaneously. The model outcomes were recalculated by running 1,000 Monte Carlo simulations. A beta distribution was used for utilities and transition probabilities, a lognormal distribution for HRs and a gamma distribution for costs. The results from the PSA were presented in cost-effectiveness planes and cost-effectiveness acceptability curves (CEAC).

### Ethic statements

In this economic evaluation study, the decision model was constructed based on data from existing literature without any direct involvement with human subjects. The BPJS Kesehatan [21] data consisted of anonymized patients record data and was publicly available. Therefore, ethical clearance was unnecessary, as the model was developed from secondary data sources.

## Results

### Base-case analysis

The base-case results, using a lifetime horizon, showed total discounted costs for trastuzumab plus chemotherapy and chemotherapy alone of US$33,744 and US$22,720 per patient, respectively, with incremental savings of US$11,024 (Table 3). Trastuzumab plus chemotherapy is associated with higher total discounted health benefits (10.10 LYs and 8.10 QALYs) compared with chemotherapy alone (8.09 LYs and 6.48 QALYs), resulting in 2.00 LYG and 1.61 QALYs

**Table 2. Model input parameters for the scenario analysis.**

| Parameters | Estimation of the base-case | 95% CI | Distribution | Source |
|---|---|---|---|---|
| **Transition probabilities** | | | | |
| Disease-free survival to locoregional recurrence | 0.017 | 0.014 to 0.020 | Beta | [3] |
| Disease-free survival to metastasis | 0.048 | 0.038 to 0.058 | Beta | [3] |
| Locoregional recurrence to metastasis | 0.048 | 0.038 to 0.058 | Beta | [3] |
| Locoregional recurrence to death | 0.012 | 0.009 to 0.014 | Beta | [3] |
| **Hazard ratio** | | | | |
| Hazard ratio overall survival | 0.610 | 0.520 to 0.730 | Log-normal | [3] |
| Hazard ratio disease-free survival | 0.580 | 0.520 to 0.660 | Log-normal | [3] |
| Hazard ratio for distant metastasis | 0.560 | 0.520 to 0.730 | Log-normal | [3] |
| **Hazard ratio disease-free survival for each age-group** | | | | |
| Age < 40 | 0.500 | 0.370 to 0.670 | Log-normal | [3] |
| Age 40–49 | 0.640 | 0.510 to 0.780 | Log-normal | [3] |
| Age 50–59 | 0.640 | 0.520 to 0.790 | Log-normal | [3] |
| Age >- 60 | 0.630 | 0.490 to 0.820 | Log-normal | [3] |

**Table 3. Deterministic result of the base-case analysis.**

| | Cost (US$) | LYG | QALY | ICER | |
|---|---|---|---|---|---|
| | | | | US$/LYG | US$/QALY |
| Chemotherapy alone | 22,720 | 8.09 | 6.48 | | |
| Trastuzumab plus chemotherapy | 33,744 | 10.10 | 8.10 | | |
| Difference | 11,024 | 2.00 | 1.61 | 5,510 | 6,842 |

Abbreviations: LYG: Life Years Gain; QALY: Quality-adjusted Life Year.

gained for trastuzumab plus chemotherapy. This resulted in an ICER of US$5,510 per LYG and US$6,842 per QALY, which is below the WTP threshold of three-time GDP.

## Scenario analysis

A scenario analysis, based on a treatment effect duration equal to the HERA trial follow-up time of 11 years, demonstrated a minimal impact on outcomes. Results show ICERs of US $5,655 per LYG and US$7,023 per QALY, which are only slightly higher than the base-case values (Table 4).

For the scenario analysis based on the joint analysis from NSABP B-31 and NCCTG N9831 [3], varying ICERs were observed among the different age groups (Table 5 and Fig 2). All scenarios resulted in ICERs below the WTP threshold. In the total population included in the joint analysis, the ICER was US$5,309 per QALY. The subgroup with an age at diagnosis <40 years had the lowest ICER of US$4,912/QALY, with an incremental gain of 3.63 QALYs and an incremental cost of US$17,853. Conversely, age at diagnosis >60 years old showed the highest ICER, with 1.80 QALYs gained, incremental cost of US$11,453, and an ICER of US$6,348 per QALY.

## Sensitivity analysis

In the one-way sensitivity analysis for the base-case analysis, the discount rate for utilities emerged as the most influential parameter in the model, followed by the discount rate for costs and the HR for DFS (Fig 3).

The results of the PSA are displayed in the cost-effectiveness plane (Fig 4). The CEAC illustrates that at a WTP threshold of three times GDP per capita, there was a 96% probability of trastuzumab plus chemotherapy being cost-effective compared to chemotherapy alone from the healthcare payer's perspective (Fig 5). The PSA for the scenario analysis based on the joint analysis from NSABP B-31 and NCCTG N9831[3] revealed that the scenario for women aged 40 or younger had the highest probability of being cost-effective, at 94% (Fig 6). Subsequently, the scenarios with age at diagnosis of 40–49 years old and 50–59 years old showed sequentially

**Table 4. Deterministic result of the scenario analysis based on an effect duration of 11-year.**

| | Cost (US$) | LYG | QALY | ICER | |
|---|---|---|---|---|---|
| | | | | US$/LYG | US$/QALY |
| Chemotherapy alone | 22,720 | 8.09 | 6.48 | | |
| Trastuzumab plus chemotherapy | 33,469 | 9.99 | 8.01 | | |
| Difference | 10,749 | 1.90 | 1.53 | 5,655 | 7,023 |

Abbreviations: LYG: Life Years Gain; QALY: Quality-adjusted Life Year.

**Table 5. Deterministic results for the scenario analysis based on a joint analysis from NSABP B-31 and NCCTG N9831 [3].**

| Intervention | Cost (US$) | Life years | QALY | ICER | |
|---|---|---|---|---|---|
| | | | | US$/LYG | US$/QALY |
| **Scenario 1 (total population included in the joint analysis)** | | | | | |
| Chemotherapy | 22,720 | 8.09 | 6.48 | | |
| Trastuzumab plus chemotherapy | 37,913 | 11.60 | 9.33 | | |
| Difference | 11,024 | 3.51 | 2.84 | 4,310 | 5,309 |
| **Scenario 2 (age at diagnosis <40)** | | | | | |
| Chemotherapy | 22,720 | 8.09 | 6.48 | | |
| Trastuzumab with chemotherapy | 40,573 | 12.58 | 10.12 | | |
| Difference | 17,853 | 4.49 | 3.63 | 3,982 | 4,912 |
| **Scenario 3 (age at diagnosis 40–49)** | | | | | |
| Chemotherapy | 22,720 | 8.09 | 6.48 | | |
| Trastuzumab with chemotherapy | 39,513 | 12.20 | 9.81 | | |
| Difference | 39,513 | 4.11 | 3.33 | 4,098 | 5,043 |
| **Scenario 4 (age at diagnosis 50–59)** | | | | | |
| Chemotherapy | 22,720 | 8.09 | 6.48 | | |
| Trastuzumab with chemotherapy | 37,599 | 11.52 | 9.27 | | |
| Difference | 14,879 | 3.43 | 2.78 | 4,342 | 5,347 |
| **Scenario 5 (age at diagnosis >60)** | | | | | |
| Chemotherapy | 22,720 | 8.09 | 6.48 | | |
| Trastuzumab with chemotherapy | 34,172 | 10.31 | 8.29 | | |
| Difference | 11,453 | 2.21 | 1.80 | 5,174 | 6,348 |

Abbreviations: LYG: Life Years Gain; QALY: Quality-adjusted Life Year.

lower probabilities of being cost-effective (93% and 91% respectively). With 82%, the scenario >60 years old had the lowest probability of being cost-effective.

## Discussion

Trastuzumab plus chemotherapy for HER2-positive early breast cancer treatment is widely used as standard of care due to its proven effectiveness in numerous clinical trials and real-world studies. Nevertheless, its application in (early) breast cancer treatment regimens in Indonesia remains limited, because its use is currently restricted to only 6 months in metastatic patients due to its relatively high acquisition cost. The findings from our cost-effectiveness analysis provide valuable insights into the economic implications of including trastuzumab in the treatment of HER2-positive early breast cancer. Our analysis shows that a one-year course of trastuzumab combined with chemotherapy is a cost-effective from the Indonesian health-care payer's perspective at a WTP threshold of three times GDP per capita. The sensitivity analysis further strengthens this conclusion and indicates a high probability of being cost-effective. Findings from the scenario analysis highlights that trastuzumab is most cost-effective when administered to individuals with an age at diagnosis below 40 years, with an ICER closely approaching the Indonesian GDP per capita.

A systematic review assessing the cost effectiveness of trastuzumab in early breast cancer in Asian countries [32] showed ICERs ranging from US$3,526 to US$56,182 per QALY. The majority of these studies took a healthcare payer's perspective and used three times GDP per capita as WTP threshold. All concluded that trastuzumab is cost-effective. Variations in ICERs observed across different studies can be attributed to country-specific differences in healthcare

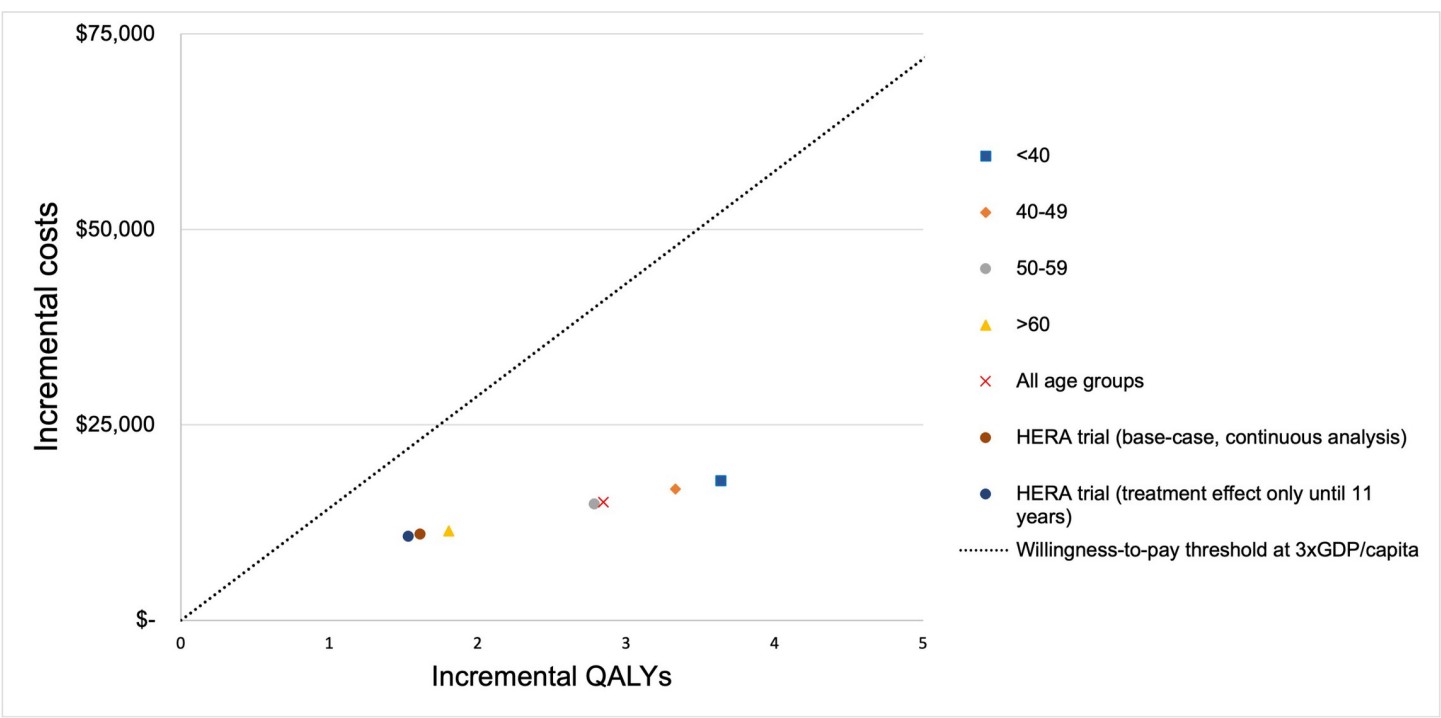

**Fig 2. Incremental cost-effectiveness ratio (ICER) for base-case based on HERA trial [5] and all scenarios[3].** Abbreviations: GDP: Gross domestic product; QALYs: Quality-adjusted life years.

systems, cost structures, modelling methodologies, and underlying assumptions. Recognizing potential heterogeneity in patient populations, healthcare settings, and resource availability is crucial and influences cost-effectiveness. This underscores the importance of conducting country-specific cost-effectiveness analyses to account for these contextual factors.

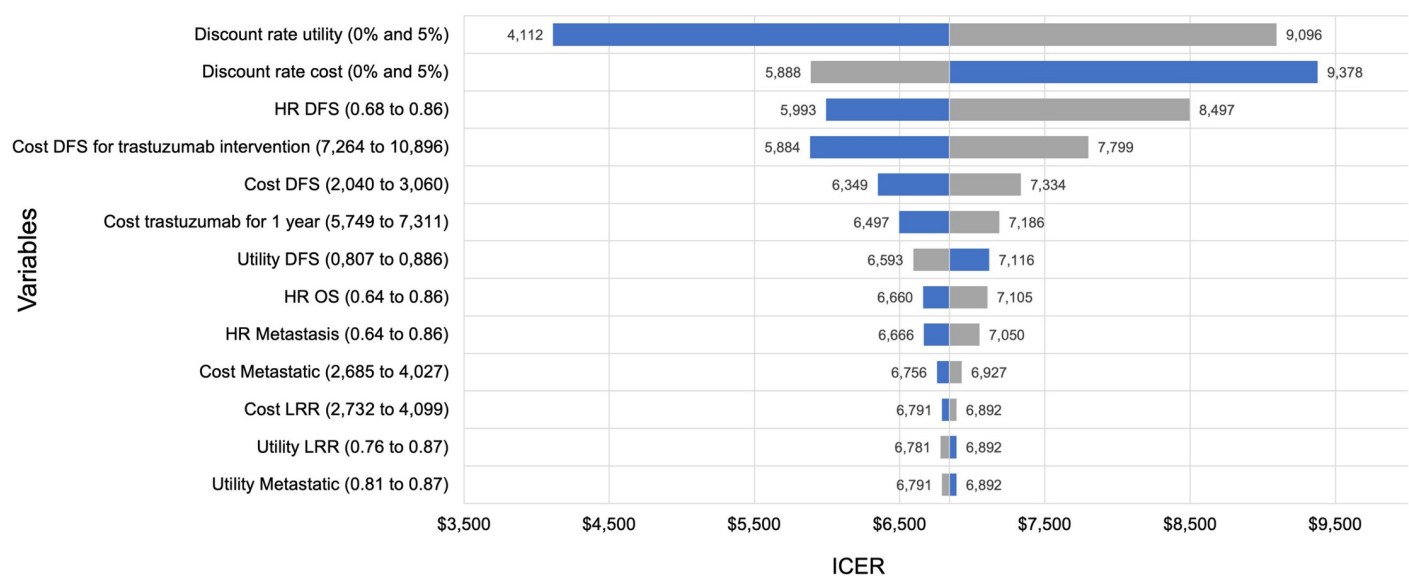

**Fig 3. Tornado diagram of one-way sensitivity analysis for the base case.** Abbreviations: DFS: Disease-free survival; HR: Hazard ratio; ICER: Incremental cost-effectiveness ratio; LRR: Locoregional recurrence; OS: Overall survival.

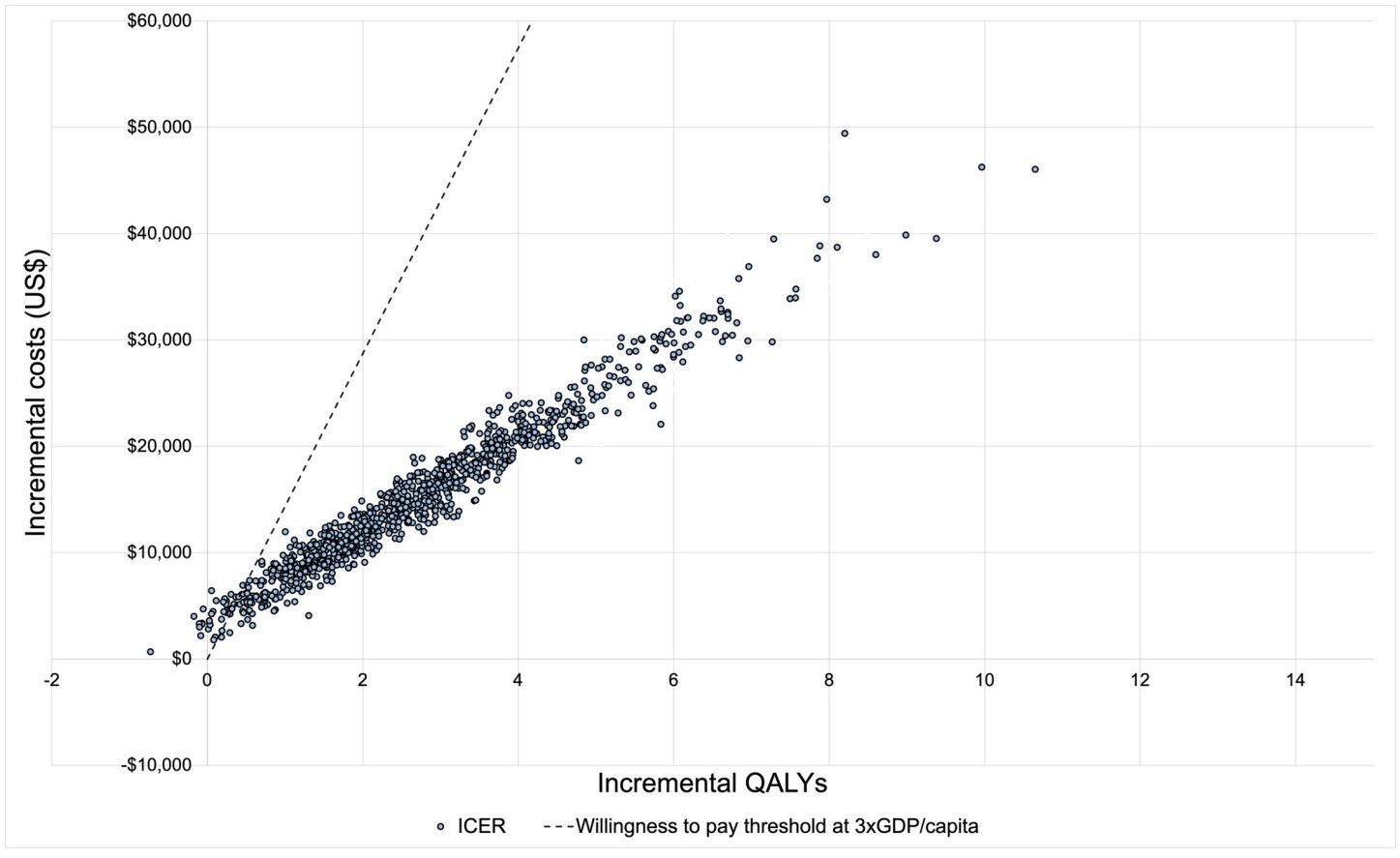

**Fig 4. Cost-effectiveness plane from the probabilistic sensitivity analysis based on 1,000 simulations.** Abbreviations: GDP: Gross domestic product; QALYs: Quality-adjusted life years.

Studies have reported a wide range of incremental LYs and QALYs respectively ranging from 0.12 to 2.87 [11–13,19,24,33] and 0.42 to 2.16 with trastuzumab treatment [12,15,20,34,35]. In our study, we identified incremental health benefits of 2.00 LYs and 1.61 QALYs with trastuzumab treatment. These values are in good agreement with the range of results reported in the existing literature, underscoring the broad consensus on the efficacy of trastuzumab in improving patient outcomes.

Our findings, based on the HERA trial [5], are consistent those reported in literature reviews [36] demonstrating the cost-effectiveness of trastuzumab based on other clinical trials such as the FinHER trial, NSABP B-31, and NCCTG N9831 trials. For instance, model-based study using healthcare perspectives found trastuzumab to be cost-effective based on data of the HERA trial and the FinHer study [11], as well as based on NSABP B-31 and NCCTG N9831 trials [37]. Conducting pharmacoeconomic evaluations is optimally achieved through the use of data originating from randomized controlled trials (RCTs), as they represent the most dependable data source with minimal biases [38] or assumptions during the data collection [39].

While extrapolating data from an 11-year clinical trial to a 50-year timeframe involves assumptions and uncertainties, it might be a justifiable approach when considering the lack of long-term empirical evidence. However, acknowledging the limitations in extrapolating data beyond the observed timeframe is important. Therefore, a scenario analysis incorporating the treatment effect of trastuzumab only for the 11-year period was conducted. Results showed

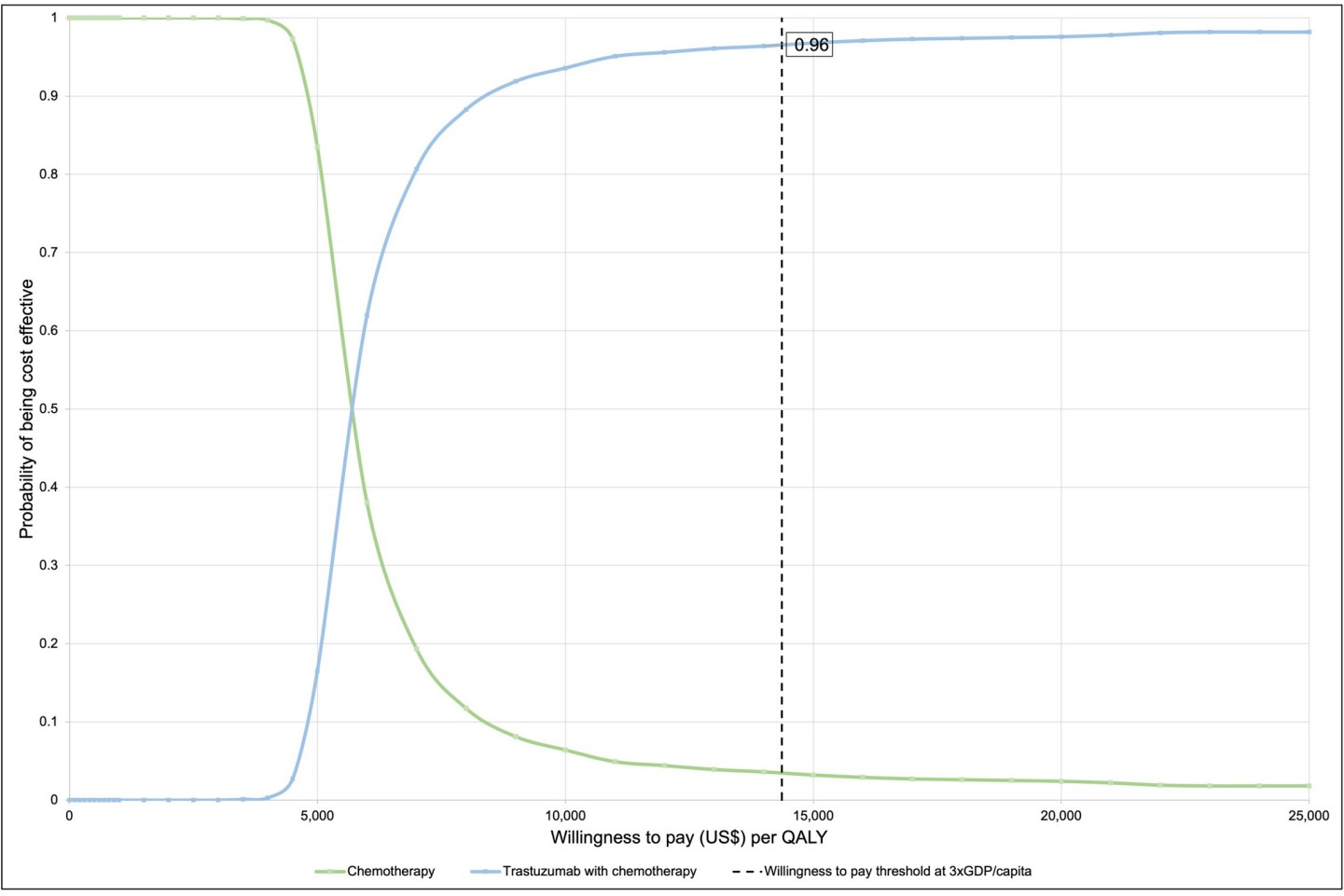

**Fig 5. Cost-effectiveness acceptability curve for the base-case analysis.** Abbreviations: GDP: Gross domestic product.

limited impact on the outcomes and did not change the conclusion that trastuzumab is cost-effective. This strengthens the robustness of our conclusion.

It is known that the effectiveness of trastuzumab treatment is dependent on the age of the patient [3]. However, no subgroup analyzes was conducted for the latest (11-year) analysis of the HERA trial. Therefore, we conducted a scenario analysis based on a joint analysis from NSABP B-31 and NCCTG N9831 [3], which also had a considerable follow-up duration and did conduct subgroup analyzes for different categories of age at diagnosis. Trastuzumab plus chemotherapy was found to be cost-effective at three times GDP in all age groups, with women aged <40 years at diagnosis being the most cost-effective. This result is in line with a similar subgroup analysis conducted in New Zealand, which showed that younger patients with breast cancer had the most favorable cost-effectiveness of trastuzumab [40]. However, starting trastuzumab administration in patients diagnosed older than 60 years old was still found to be cost-effective. Some studies show that trastuzumab was cost effective for women younger than 70, but not in older patients [40,41]. However, considering the most recent evidence of effectiveness and the availability of trastuzumab biosimilars (price one-third lower than branded Herceptin®), the likelihood of trastuzumab being cost effective is expected to increase, particularly for those with poorer prognosis, but also in the older population [42].

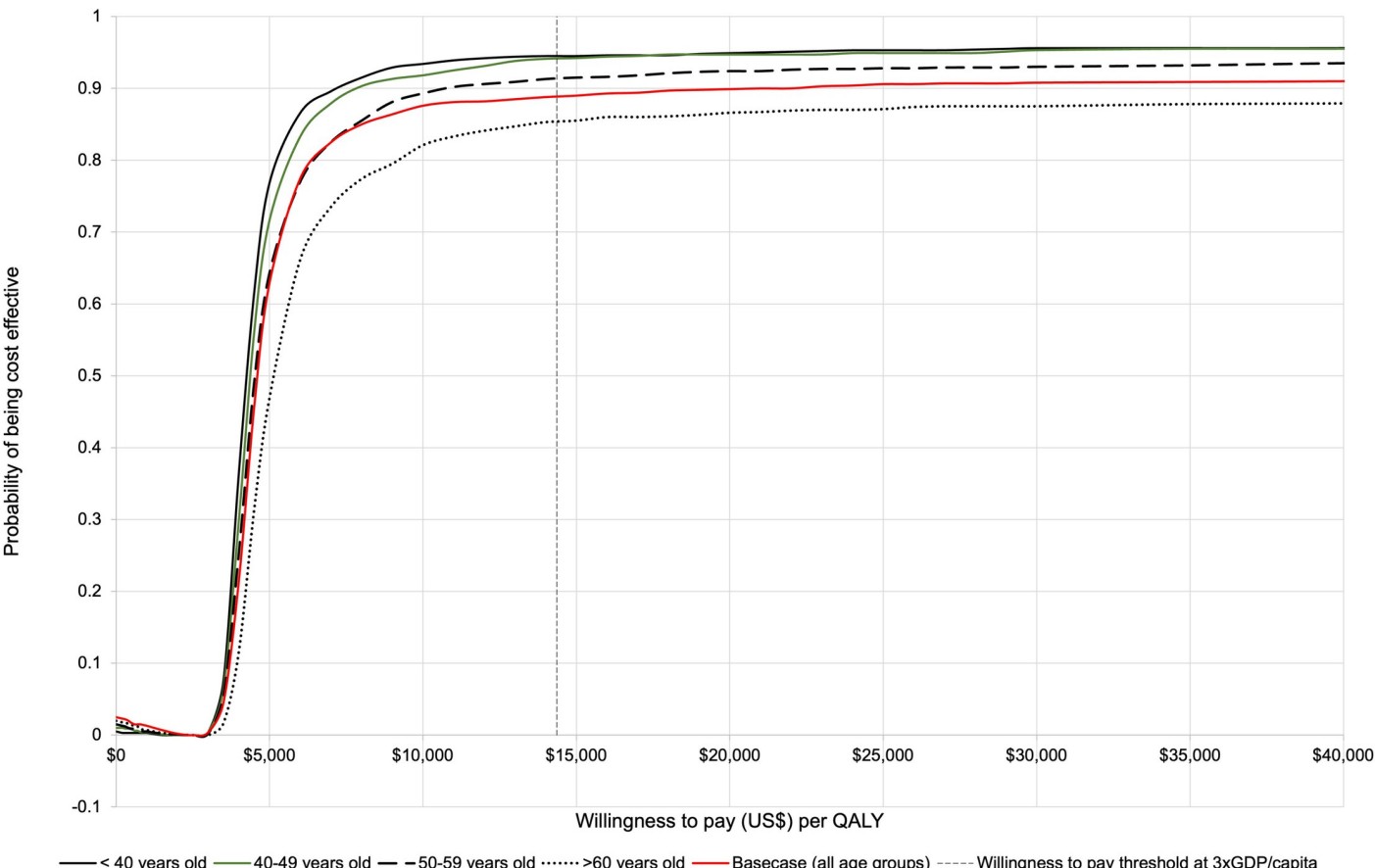

**Fig 6. Cost-effectiveness acceptability curve for the scenario analysis based on a joint analysis from NSABP B-31 and NCCTG N9831[3].** Abbreviations: GDP: Gross domestic product; QALYs: Quality-adjusted life years.

It is crucial to explicitly clarify that our results are intended to illustrate the variability of cost effectiveness with respect to different parameters, rather than serving as definitive guidelines for treatment allocation. Oncologists are tasked with considering patient-specific factors, including comorbidities, patient preferences, potential toxicities, and other pertinent variables. Our analysis adds to the growing body of evidence showing how incorporating heterogeneity in cost-utility analysis can facilitate more informed decision-making. Ignoring this heterogeneity poses the risk obscuring the fact that an intervention that is considered cost effective on average for a given population, may not apply to specific subgroups within that population (e.g., based on age). Consequently, such oversight could result in a suboptimal allocation of resources.

One of the notable strengths of our analysis is the use of clinical data from the 11-year follow-up of the HERA trials, while most cost-effectiveness analyzes of trastuzumab rely on shorter-term data from the HERA trials. Second, by incorporating scenario analyzes based on age groups, we shed light on the potential variation in cost-effectiveness across different patient subgroups. This underscores the importance of personalized medicine and ensuring that resource allocation should be based on each patient's specific needs and characteristics.

As no clinical trials on trastuzumab in HER2-positive early breast cancer were conducted in Indonesia, there is no data available regarding the drug's efficacy specific for the Indonesian population. Therefore, it was assumed that the findings from the clinical trials (HERA and

joint analysis) can also be extrapolated to Indonesia. Additionally, because of limitations in available data, the resource use for disease management in each health state was collected from a published study [24]. Moreover, this analysis does not consider the societal perspective. A societal perspective could capture a more comprehensive assessment by including productivity losses and informal care. However, due to data constraints, it was not possible to consider this perspective. It is expected that the ICER from a societal perspective could potentially be even lower, considering additional prevented societal costs such as productivity losses. An additional limitation concerns the exclusion of trastuzumab-related cardiotoxicity from our model. Although cardiotoxicity is a reversible condition that does not lead to increased mortality, it may still have a modest impact on quality of life estimates. However, given the low incidence [43] and based on sensitivity analyzes conducted in previous economic evaluations, this is expected to have only a minor impact on the overall cost-effectiveness results.

Further research and ongoing evaluation are warranted to account for evolving clinical practices, emerging therapies, and changes in healthcare systems, which may impact the cost-effectiveness of trastuzumab over time, particularly by utilizing real-world evidence data specific to the Indonesian context. Ideally, an evaluation should be conducted to compare real-world evidence with clinical trial findings, to gain valuable insights into the effectiveness and cost-effectiveness of trastuzumab treatment in Indonesia. Moreover, additional exploration of heterogeneity can be conducted by considering factors such as the number of hormone receptor status [40], tumor size/grade [44], and cardiovascular risk factors [12]. While our analysis incorporated the most up-to-date efficacy data from clinical trials and supplemented it with reimbursement cost data in Indonesia, it is worth noting that several trials evaluating shorter regimens are anticipated to release their results. Conducting a meta-analysis of these trials will provide perspectives for determining the optimal duration of trastuzumab treatment and the durability of its effectiveness.

## Conclusions

Although addition of trastuzumab to chemotherapy in HER2-positive breast cancer patients entails additional treatment costs, our analysis suggests that its use is justifiable from a healthcare payer's perspective. Trastuzumab plus chemotherapy can be considered cost-effective in Indonesian breast cancer patients compared with standard chemotherapy alone and is likely to be most cost-effective in women <40 years of age.

## Supporting information

**S1 File. Supplementary material.**
(DOCX)

## Author Contributions

**Conceptualization:** Sudewi Mukaromah Khoirunnisa.

**Formal analysis:** Sudewi Mukaromah Khoirunnisa.

**Investigation:** Sudewi Mukaromah Khoirunnisa, Fithria Dyah Ayu Suryanegara, Lisa Aniek de Jong.

**Methodology:** Sudewi Mukaromah Khoirunnisa, Fithria Dyah Ayu Suryanegara, Didik Setiawan, Lisa Aniek de Jong.

**Supervision:** Didik Setiawan, Maarten Jacobus Postma, Lisa Aniek de Jong.

**Validation:** Lisa Aniek de Jong.

**Writing – original draft:** Sudewi Mukaromah Khoirunnisa.

**Writing – review & editing:** Sudewi Mukaromah Khoirunnisa,
Fithria Dyah Ayu Suryanegara, Didik Setiawan, Maarten Jacobus Postma,
Lisa Aniek de Jong.

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
