## [Decision Letter · Decision Letter 0]

2 Jan 2024

PONE-D-23-33455Economic evaluation of trastuzumab in HER2-positive early breast cancer in Indonesia: a cost-effectiveness analysisPLOS ONE

Dear Dr. khoirunnisa,

Thank you for submitting your manuscript to PLOS ONE. After careful consideration, we feel that it has merit but does not fully meet PLOS ONE’s publication criteria as it currently stands. Therefore, we invite you to submit a revised version of the manuscript that addresses the points raised during the review process.

Kind regards,

Nabil Elhadi Elsayed Ali Omar, PharmD, BCOP, PhD (C)

Academic Editor

PLOS ONE

A clean copy of the edited manuscript (uploaded as the new *manuscript* file).

“the Indonesia Endowment Funds for Education (LPDP).”

6. We note that your Data Availability Statement is currently as follows: [All relevant data are within the manuscript and its Supporting Information files.]

7. Please amend either the abstract on the online submission form (via Edit Submission) or the abstract in the manuscript so that they are identical.

Reviewers' comments:

Reviewer's Responses to Questions

**Comments to the Author**

1. Is the manuscript technically sound, and do the data support the conclusions?

Reviewer #1: Yes

Reviewer #2: Yes

2. Has the statistical analysis been performed appropriately and rigorously? 

Reviewer #1: Yes

Reviewer #2: Yes

3. Have the authors made all data underlying the findings in their manuscript fully available?

Reviewer #1: Yes

Reviewer #2: Yes

4. Is the manuscript presented in an intelligible fashion and written in standard English?

Reviewer #1: Yes

Reviewer #2: Yes

5. Review Comments to the Author

Reviewer #1: The manuscript is very interesting and well written, although I believe these comments could improve your manuscript.

The abstract and introduction are interesting and well written.

Materials and Methods

Table 1 and Table 2; Hazard ratio overall survival, Hazard ratio Disease-free survival, and Hazard ratio for distant metastasis in table 1 from HERA trial, and table 2 from the joint analysis of NSABP B-31 and NCCTG177 N98313, why did you use table 1 inputs for base case with ICER 6,842 per QALY then use table 2 input for total population included in the joint analysis with ICER was US$5,292 per QALY, however you could use the table 2 inputs for both base case and Scenario analysis ( age subgroup), please clarify the value of using both data as it could be confusing for the reader. Note extracting HR for general population from a meta-analysis for the general population will give you more robust results instead of using 2 different RCT respecting to the recommendation of evidence ranking in health economics ( less confusing and more consistent), there are a meta-analysis 2021 covering this topic

Line 195 “per-capita GDP” please add the value of per capita GDP in table 1 and illustrate per capita GDP for which year.

Line 241 “For the total population included in the joint analysis, the ICER was US$5,292 per QALY.” It is not clear how this ICER was calculated (depends on which HR)

Line 323 HERA trial you need to add the reference here.

Line 158 “For estimating costs per health states, the proportion of resources used for each health states was utilized to finally calculate the total cost annually” as you depended on procedures costs from another country (India), it would be better to mention that in the limitation, as economic evaluations depend mainly on the costs in your country to be freely compare the ICER to your WTP threshold.

Results are well interpreted , although it is recommended to improve the resolution of figure 2,3,4,5,6.

The discussion is well written.

Very important topic but you need to clarify a few points to be more reliable and reproducible

Reviewer #2: Can you please provide a rational for extrapolating the transition probabilities and HRs from the mentioned clinical trials (follow up period 11 years) to 50 years in the Markov model? What suggests they will remain unchanged after 11 years?

6. PLOS authors have the option to publish the peer review history of their article (what does this mean?). If published, this will include your full peer review and any attached files.

Reviewer #1: **Yes: **Shaimaa Abdelaziz Abdelmoneim

Reviewer #2: No

---

## [Author Response · Author response to Decision Letter 0]

15 Mar 2024

Reviewer 1

1. Materials and Methods

Table 1 and Table 2; Hazard ratio overall survival, Hazard ratio Disease-free survival, and Hazard ratio for distant metastasis in table 1 from HERA trial, and table 2 from the joint analysis of NSABP B-31 and NCCTG177 N98313, why did you use table 1 inputs for base case with ICER 6,842 per QALY then use table 2 input for total population included in the joint analysis with ICER was US$5,292 per QALY, however you could use the table 2 inputs for both base case and Scenario analysis ( age subgroup), please clarify the value of using both data as it could be confusing for the reader. 

Note extracting HR for general population from a meta-analysis for the general population will give you more robust results instead of using 2 different RCT respecting to the recommendation of evidence ranking in health economics (less confusing and more consistent), there are a meta-analysis 2021 covering this topic

Respond: The ICER for the base case was based on input parameters in the Table 1 (HERA trial), while in scenario analysis, the ICER for total population and age group in scenario analysis, was based on the joint analysis of NSABP B-31 and NCCTG177 N98313. The HERA trial was used in the base case analysis as this is the longest data available for the use of trastuzumab in breast cancer. We clarified about this in the the Materials and Methods, subsection Transition probabilities, Line 137-138.

Thank you for your thoughtful suggestion regarding the extraction of hazard ratios (HR) for the general population from a meta-analysis. We acknowledge the importance of adhering to the recommended evidence ranking in health economics for a more robust and consistent approach. In response to your recommendation, we have explored a meta-analysis conducted in 2021 that provides a comprehensive analysis of the effectiveness and outcomes associated with trastuzumab treatment in breast cancer patients. However, the meta-analysis emphasized the analysis of the long-term benefits and risks of adjuvant trastuzumab on breast cancer recurrence and cause-specific mortality, which are not relevant to our model.

2. Line 195 “per-capita GDP” please add the value of per capita GDP in table 1 and illustrate per capita GDP for which year.

Respond: 

Thank you for your comment. The value of per capita GDP was added in Table 1.

3. Line 241 “For the total population included in the joint analysis, the ICER was US$5,292 per QALY.” It is not clear how this ICER was calculated (depends on which HR)

Respond: Thank you for your clarification. The explanation of the ICER calculation for the total population for scenario analysis was mentioned now in the Materials and Methods, subsection Scenario analysis, Lines 182-184.

4. Line 323 HERA trial you need to add the reference here.

Respond: 

Thank you for your comment. The reference (Line 304) was added. 

5. Line 158 “For estimating costs per health states, the proportion of resources used for each health states was utilized to finally calculate the total cost annually” as you depended on procedures costs from another country (India), it would be better to mention that in the limitation, as economic evaluations depend mainly on the costs in your country to be freely compare the ICER to your WTP threshold.

Respond: 

Thank you for your comments. 

The procedure costs used in this study were derived from BPJS Kesehatan (mentioned in Materials and Methods, subsection Utilities and costs, Lines 149-154). However, due to the data constraints related to costs for each health states in Indonesia, we used the proportions of the patients receiving the treatments for each health states from India to finally calculate the health states costs. The limitation of the use was now mentioned in Discussion, Lines 353-355.

6. Results are well interpreted, although it is recommended to improve the resolution of figure 2,3,4,5,6.

Respond: 

Thank you for your suggestions. We have improved the quality of the figures.

7. The discussion is well written.

Very important topic but you need to clarify a few points to be more reliable and reproducible

Respond: 

Thank you for your feedback.

We greatly appreciate your positive assessment of the discussion section and acknowledge your valuable input regarding the need for clarification on certain points to enhance the reliability and reproducibility of our study.

We have thoroughly reviewed your comments and have taken the necessary steps to address them in our manuscript. We believe that these revisions have significantly strengthened the reliability and reproducibility of our findings.

Once again, thank you for your thoughtful feedback. We are confident that the improvements made will enhance the quality of our manuscript.

Reviewer 2

Can you please provide a rational for extrapolating the transition probabilities and HRs from the mentioned clinical trials (follow up period 11 years) to 50 years in the Markov model? What suggests they will remain unchanged after 11 years?

Respond: 

Thank you for your comments. While extrapolating data from an 11-year clinical trial to a 50-year timeframe involves assumptions and uncertainties, it might be a justifiable approach when considering the lack of long-term empirical evidence. However, acknowledging the limitations and uncertainties associated with this extrapolation is essential, and conducting sensitivity analyses and seeking expert input can enhance the credibility of these long-term predictions within the Markov model.

In response to this comment, we conducted an additional sensitivity analysis covering this limitation, by simulating a treatment effect duration of trastuzumab of 11 years (equal to the follow-up of the trial) (Methods, subsection Scenario analysis, Lines 117-180). The ICER result based on the scenario analysis incorporating the efficacy of trastuzumab until 11 years showed that trastuzumab is cost-effective (Results, subsection Scenario analysis, Lines 223-225. We have now also added a paragraph in the discussion about this (Discussion, Lines 312-318)

---

## [Decision Letter · Decision Letter 1]

14 May 2024

Economic evaluation of trastuzumab in HER2-positive early breast cancer in Indonesia: a cost-effectiveness analysis

PONE-D-23-33455R1

Dear Dr. khoirunnisa,

We’re pleased to inform you that your manuscript has been judged scientifically suitable for publication and will be formally accepted for publication once it meets all outstanding technical requirements.

Kind regards,

Nabil Elhadi Elsayed Ali Omar, PharmD.,BCOP.,PhD(C)

Academic Editor

PLOS ONE

Additional Editor Comments (optional):

Reviewers' comments:

Reviewer's Responses to Questions

**Comments to the Author**

1. If the authors have adequately addressed your comments raised in a previous round of review and you feel that this manuscript is now acceptable for publication, you may indicate that here to bypass the “Comments to the Author” section, enter your conflict of interest statement in the “Confidential to Editor” section, and submit your "Accept" recommendation.

Reviewer #2: All comments have been addressed

2. Is the manuscript technically sound, and do the data support the conclusions?

Reviewer #2: Yes

3. Has the statistical analysis been performed appropriately and rigorously? 

Reviewer #2: Yes

4. Have the authors made all data underlying the findings in their manuscript fully available?

Reviewer #2: Yes

5. Is the manuscript presented in an intelligible fashion and written in standard English?

Reviewer #2: Yes

6. Review Comments to the Author

Reviewer #2: (No Response)

7. PLOS authors have the option to publish the peer review history of their article (what does this mean?). If published, this will include your full peer review and any attached files.

Reviewer #2: No

---

## [Editor Report · Acceptance letter]

16 May 2024

PONE-D-23-33455R1 

PLOS ONE

Dear Dr. Khoirunnisa, 

I'm pleased to inform you that your manuscript has been deemed suitable for publication in PLOS ONE. Congratulations! Your manuscript is now being handed over to our production team.

Kind regards, 

on behalf of

Dr. Nabil Elhadi Elsayed Ali Omar 

Academic Editor

PLOS ONE